# The Need for a Food Systems Approach in Smallholder Food and Nutrition Security Initiatives: Lessons from Inclusive Agribusiness in Smallholder Communities

**DOI:** 10.3390/foods10081785

**Published:** 2021-07-31

**Authors:** James Wangu

**Affiliations:** Department of Human Geography and Spatial Planning, Utrecht University, 3584 CB Utrecht, The Netherlands; j.m.wangu@uu.nl

**Keywords:** food systems, agribusiness, non-food needs, household priorities, food security, rural diversity

## Abstract

Inclusive business models dominate current development policy and practices aimed at addressing food and nutrition insecurity among smallholder farmers. Through inclusive agribusiness, smallholder food security is presumed to come from increased farm productivity (food availability) and income (food access). Based on recent research, the focus of impact assessments of inclusive business models has been limited to instrumental aspects, such as the number of farmers supported, the training provided, and immediate farm outcomes, namely revenue. Furthermore, the assessments limit their scope to participating smallholders, while overlooking other community members. With respect to food and nutrition security, there is no acknowledgement of the diverse household needs that compete with the food requirements with regard to the multi-dimensional nature of poverty. Focusing on recent studies and reviews on the contribution of inclusive business initiatives to smallholders’ livelihoods and food security, the present review adopts a food systems approach for broader knowledge and insight analysis. It re-emphasizes that a food systems approach that provides a systemic and broader way of thinking about and working on food issues is critical for development initiatives aimed at ensuring that every person can meet their food and nutrition needs.

## 1. Introduction

Over the last decade, inclusive business models (IBMs) have dominated development policy and practices aimed at improving low-income smallholder farmers’ food and nutrition security—hereafter referred to as food security [1,2]. Generally, IBMs are defined as commercially viable business initiatives that integrate low-income populations into their value chains as suppliers, processors, distributors, retailers, or consumers [3,4,5,6]. Popular, and preferred over traditional development approaches such as aid and charity, the strength of the inclusive business approach is in the business models’ potential to achieve commercial success while addressing social challenges [1,7]. Among smallholders, it is maintained that in the adoption of IBMs, private sector actors stand to gain from enormous opportunities in food value chains, in terms of production, distribution, and consumption, from enhanced smallholder agri-food sector growth, whereas food-insecure smallholders would benefit from improved capacity to address production constraints such as inputs, extension services, technology, and marketing [5,7,8,9,10]. Through IBMs, it is presumed that smallholder food security would result from enhanced farm productivity (food availability) and income (food access). Selected literature on the inclusive business approach, however, has expressed reservations about whether such a positive contribution can be effectively realized; others maintain that the approach may lead to negative outcomes [11,12,13]. The following contexts highlight the different areas of concern. First, due to competitiveness in the agri-food industry, IBMs may not adequately address constraints facing smallholder farming, leading to limited livelihood changes and/or exclusion of some smallholders in the businesses [8,14,15,16]. Second, for the businesses’ participants, the increase in farm productivity and income may not be sufficient to counter their households’ food insecurity challenges. Participation in the businesses implies market-oriented production instead of for-home consumption; however, the majority of food consumed in smallholder households comes from personal production [17,18,19,20]. Increased focus on the market requires more investment from the farmers to meet the production costs and is likely to result in dependence on the market for food. It may also expose smallholders to food price fluctuations [21,22,23]. Furthermore, the level of improvement in smallholders’ farm incomes is a significant determinant of their food security. Smallholders are not only among the most food-insecure population in the world, but are also among the most poor (reference). Due to multi-dimensional nature of poverty, the income derived from the business is likely to be utilized for multiple households’ needs, including health, education, agricultural inputs, and home improvement, in addition to meeting food needs. Therefore, the effect of increased farm income on food security depends on the amount of money made and the priorities accorded to the different households’ needs. Understanding this complexity in the relationship between IBMs and smallholder household food security outcome requires a broader lens rather than the productivity and income perspective used in the inclusive business approach.

In this paper, a food systems approach is suggested for understanding the extent to which IBMs can contribute to household food security in smallholder communities. A food systems approach provides for ‘a more holistic way of thinking and working’ toward ensuring that every person can meet their food and nutrition needs [24]. The approach ensures acknowledgement of the intricate nature of the interaction between the elements (natural, technical, economic, social, and political) and activities central to food production, processing, distribution, consumption, and disposal [24,25,26]. In the food systems’ approach, the delivery of food is viewed as a complex interaction of sub-systems with a feedback loop rather than a simple, linear process [27,28,29]. Thus far, the available literature on food systems has focused on global food security (more generally), health, biodiversity, environmental impact, and a combination of these aspects vis-à-vis food security [27,29,30,31,32,33,34,35,36,37].

Utilizing the food systems approach, the available empirical inquiries into the (potential) contribution of inclusive business initiatives to smallholders’ livelihoods (food security) is re-analyzed. Included in the analysis are three case studies that are part of a recently concluded research project—Follow the Food. The project assessed the contribution of foreign-induced agribusiness investments to local food security [38]. This author contributed primarily to two of these studies: the French bean and mango cases. [39,40]. The Malt barley case constitutes the third study from Follow the Food [41]. For the rest of the empirical cases, a search was conducted using Google Scholar through the Utrecht University library, with the following keywords: inclusive; (agri)business; food and nutrition security; smallholder. All studies from last decade (2010–2021) on inclusive business model(s) in Africa were included in this review. To broaden the scope of the analysis, two review publications on inclusive business in smallholder communities were also included. In total, 10 empirical studies, two review articles, and two organization reports were analyzed/reviewed. Table 1 provides an overview of the materials. Further detail on the initiatives is presented in following sections.

From the Follow the Food cases, it was evident that there are significant limitations to using IBMs as the solution to food security in smallholder farming communities. Generally, these limitations arise from the mismatch between the initiatives and local priorities and context as well as the instrumental focus of initiatives—farm productivity and income—instead of the desired end goal—food security. By re-analyzing the findings of these studies from a food systems perspective, whereby food is the center of focus, the systemic issues surrounding the effect of IBMs on local food security (or lack thereof) are better understood. Guiding the present analysis are two key questions. One, are all smallholders in the targeted communities, especially the most food insecure, able to derive farm benefits from inclusive business initiatives? Two, do inclusive business initiatives and resulting income levels sufficiently address the local non-farming-related food access hurdles? Smallholder farming systems, characterized by a range of constraints including limited access to land and water, as well as farm inputs and technology, are central to food availability and access in smallholder communities. Inclusive businesses must, therefore, effectively counter these systemic constraints to boost local food availability and/or access. In doing so, they must consider the diversity in constraints common in different households within the same community. The food availability dimension is determined by the type of produce promoted through the IBMs and the destined markets. Locally consumed food, which is destined for local markets, would arguably have a more favorable effect on the local food system than food destined for distant markets. To enhance smallholder food security via food access, inclusive business initiatives and anticipated improvement in income must counter socioeconomic (system) barriers to food access in respective communities. These barriers include the aforementioned competing non-food needs (healthcare, school fees, clothing, etc.), poor local food market conditions, and limited nutrition education. Prior to delving into the analysis and detailed findings, a background of the food systems approach is provided.

## 2. The Food Systems Approach—A Conceptual Background

Today, policymakers and major international development agencies, such as the European Commission and the Food and Agricultural Organizations (FAO), are working through the food systems approach in their global food security efforts, as is clearly evidenced in various high-profile food and nutrition security reports [34,53,54]. This years’ United Nations Food Systems Summit (2021) is a testament to the growing recognition of the systems approach to working on global food security [55]. As Halberg [56] suggests, the food systems concept is perceived as an alternative to ‘value chains as the common description of food production from field to fork’. Generally, value chain analysis entails mapping actors and key linkages within a chain, whereby the focus is establishing areas of improvement via, among other methods, better governance [57,58,59]. The value chain analysis is critical to food security, as it informs food systems’ assessment [60,61,62,63]. However, the value chain approaches are perceived as limited for undertaking a broader and integrated analysis of food systems. Integrated analysis allows for ‘consideration of multiple outcomes, and better links value chains with consumer behaviour and food environments’ [64,65]. Figure 1 is a graphical representation that shows the differences and similarities between the food systems framework (on the left) and the value chain framework.

Food systems comprises interactions between bio(geo)physical and social, economic, and political factors [37,68,69]. The analysis of these interactions requires a food system to be perceived as comprising ‘determinants and outcomes of its activities’ [70]. The determinants constitute the biophysical and the social, economic, and political elements that guide the performance of the system’s activities. The outcomes give rise to a certain level of food security but could also result in social and environmental changes that in turn influence the food system activities and its outcomes [70,71,72,73,74].

The concept of food systems is not new, having its roots in the 1970s and 1980s [75,76]. Recent renewed attention on the concept among policymakers and in research and practice has emerged from a need to understand the increasingly globalized food system with respect to concerns over how food systems’ structure and governance impact the patterns of consumption and social and environmental welfare [37,68,77,78]. In light of this, there is a consensus that food systems are failing, given the mounting problem of hunger and malnutrition across the globe; thus, there is an urgent need for their improvement [36,79,80,81]. However, ‘the nature of what that failure actually entails seems to differ considerably between perspectives’ [68].

Food systems are complex, owing to the many interactions involved; thus, adopting the food systems approach in research and practice is a challenge. In smallholder communities, for instance, local food systems are less structured/formal, making them difficult to adequately analyze. Adding to the complexity, the FAO [65] insists a food systems review should consider contemporary global changes such as population growth, natural resource depletion, climate change, and globalization. One major weakness is the fact that even though the approach offers a broad and integrated way of analyzing food security issues in specific communities, it does not offer guidance on the most important areas to focus on. In principle, food systems are boundless. Notwithstanding its shortcomings, i.e., a lack of focus and delineation compared to the value chain approach commonly employed to study IBMs, it promises to provide a broader—namely, systemic—understanding of the critical factors that determine food security outcomes in smallholder communities.

Research and policy on food systems has revolved around three areas: agriculture, nutrition, and (socio-)ecology. In the agriculture domain, attention is narrowly focused; the emphasis is on ‘maintaining or restoring productivity’, especially in the context of climate change [68]. Other concerns entail the negative effect food systems may have on the environment, such as resource depletion [82]. Limited attention has been given to the nutrition domain. Efforts have been focused on addressing the poor quality of diets and eating habits through supplementation and food fortification campaigns and encouraging dietary behavioral changes such as breastfeeding [68,83,84]. Yet, the long-term efficacy of such strategies has been questioned in the event that they are implemented outside the broader local food environment and when they do not address the root causes of undernutrition [84,85]. The (socio-)ecology context focuses on the interactions between bio-physical and social dynamics, and how these components are impacted by environmental, social, and livelihood factors [68].

In analyzing the contribution of IBMs to smallholder household food security, all three areas are relevant. Low productivity (and as a result, low income), which characterizes smallholder farming, is perhaps the primary reason the local food systems fail to deliver adequate and nutritious food. Therefore, IBMs’ promise to address the contributing constraints, i.e., improving smallholder farming systems, is a critical step towards local food security. However, the anticipated outcomes from IBMs, namely increased productivity and income, are contingent on the nature and the scope of the inclusivity aspect, the agricultural and innovation systems, which are among the food system’s sub-systems. Other local sub-systems are also significant [68,77,86,87]. Among these sub-systems are the socio-ecological systems that revolve around quantity and quality of land and water resources, cultural context (food preferences), and political-ecology systems that determine issues of politics, power, and social justice, which are critical to food security for vulnerable households. Overall, how inclusive business initiatives interact and affect the different sub-systems’ performance determines the ultimate local food security outcome.

The present analysis begins with a review of the scope of different IBMs in different communities, as specified in the selected studies. The results provide answers to the first question of this paper—whether all farmers in targeted communities derive benefit from IBMs—by establishing who they are and how they are integrated. Specifically, it addresses whether all farmers in targeted communities are integrated into the IBMs as well as the quality of support they receive vis-à-vis addressing individual farming constraints.

## 3. The Scope of Inclusive Business Models

### 3.1. Description of the Initiatives

The United Nations World Food Programme (WFP) maintains that boosting smallholders’ livelihood by providing them with support to tackle local production constraints and limited access to (formal) markets is ‘key to building sustainable food systems, advancing food security and achieving Zero Hunger’ [88]. This is precisely what inclusive business models in smallholder communities seek to accomplish. The malt barley initiative that aimed to address food insecurity in the Lay Gayint district in northern Ethiopia took the form of contract arrangements between Gondar Malt Factory (GMF) and smallholders’ cooperatives [42]. Inclusion elements from the business comprised a guaranteed produce market as well as access to credit, technology, and technical skills. A state-sponsored vegetable initiative in the Raya Azebo district, also located in northern Ethiopia, promoted commercial vegetable production (hot peppers, onions, and tomatoes) for local, regional and, occasionally, export markets (Djibouti) [43]. A shift from less profitable staple food production to horticultural crops for higher returns is considered a key pathway to local food security (increasing food access). The inclusivity aspects of the initiative include irrigation infrastructure, provision of extension services, and facilitation to reach produce markets. Community Revenue Enhancement through Agricultural Technology Extension (CREATE) is another initiative (Public Private Partnership (PPP)) in Ethiopia’s, Arsi Zone, having a similar objective to the first two initiatives. It is financed by the Dutch government, Heineken, and the World Bank and involves malt barley production [41]. Heineken provides the market for the mart barley, which is used to produce beer locally. Through contract farming, smallholders have access to quality inputs, particularly, new, high-quality seed varieties, training, extension services and a stable, guaranteed market for their produce. Finally, six different inclusive business initiatives in Ethiopia, seeking to address local food security, are researched in one case study [44]. These include (1) spice and grain initiative—a PPP arrangement that supports smallholders’ access to quality inputs, technical skills, technology, and markets; (2) a PPP initiative that promotes the avocado value chain; (3) a PPP promoting coffee production and marketing; (4) a daily initiative by a foreign company supporting smallholders with quality animal feeds, essential hardware, and transport facilities as a means to integrate them into the dairy value chain; (5) a livestock PPP initiative by a private company that links smallholders to national and export markets for meat products; (6) a PPP initiative that promotes seed production by smallholders by equipping them with the right inputs, information, and technology.

Elsewhere, in northern Ghana, a farmers’ association, Masara N’arziki, established by two partly foreign and private agribusiness companies, promotes commercialization of various commodities, including cocoa, rice, maize, and cotton [45]. The study included here focuses on maize. The business supports smallholders through subsidized agricultural inputs, services, and linkage to markets. The other case from Ghana involves an IBM of a company, HPW Fresh & Dry Ltd., (Accra, Ghana) which promotes the production and marketing of dried fruits (pineapples, mangoes, coconuts, and papaya) for European markets [46]. The initiative provides training on resource management, imparting technical skills to farmers to improve their production capacity while providing other farm- and market-related support.

In Tharaka Nithi county in Kenya, the French bean PPP initiative, funded by the Dutch government and a private Kenyan company, introduced the production of a high-value crop—French beans—via a contract farming arrangement. It was conceived as an ideal strategy to counter dwindling farm plot sizes and provide a solution to local food insecurity in a densely populated smallholder community [39]. Through this initiative, farmers are provided access to farm inputs in the form of seeds, agrochemicals and fertilizers, and extension services and markets. The second case is from Makueni county in Kenya. The program, focused on mango production, value addition, and processing is led by the county government and funded by the European Union. The initiative follows the establishment of a fruit processing plant in the community. The goal is to solve the problem of (mango) post-harvest loss, a common experience in the community, thereby improving smallholder income and food security accordingly.

The Agro Initiative Zimbabwe (AIZ), implemented in Honde Valley and the Mutasa district of Manicaland, involves the production and marketing of tabasco chili through contract farming. Farmers receive support to access inputs, are trained in production skills, and are provided access to ready markets for their produce. The case from Tanzania includes various initiatives promoting contract farming and cooperative arrangement as ways to link farmers to the market. Among the crops produced are cashew nuts, coffee, cotton, tobacco, sesame, and pigeon peas.

In a review of aquaculture development in low- and middle-income countries, the strategies for inclusive business are similar to those described in the above cases [49]. Through contract farming, smallholders’ capacities are improved through provision of farm inputs, technical training, and market linkage, achieved through farmers’ cooperatives, associations, and groups. The study by German et al. [50] specifically reviews ‘the structural factors shaping agricultural value chains and their implications for social inclusion’. The final case study included, a farm forestry project by the International Financial Cooperation (IFC) in a ‘large South Asia country’, promoted the supply of pulpwood (eucalyptus trees) for the local pulp mills. It was geared towards enhancing local livelihoods, with the IFC supporting low-income smallholders with skills training, inputs, and credit to boost farmers’ earnings in order to alleviate their poverty and improve food security [51,52].

### 3.2. Inclusion and Quality of Inclusion

Common in all the so-called ‘inclusive’ business models in the studies assessing smallholders’ participation is the selective nature of who participates. This is to be expected, as not everyone in the respective communities can and/or is willing to be part of the initiatives because of various reasons. However, it is consistently evident that the relatively well-off farmers seem to be favored by the opportunities the businesses provide. Gebru et al. [43] find that while the vegetable initiative in Ethiopia led to increased household income, participation was largely determined by access to production resources. These include farm plot size and livestock holding, access to extension services, and the ability to meet production costs. Other (social) factors involved are cooperative memberships, market risk perception, price of the produce, and distance to the market. These findings are mirrored to a certain extent by the French bean initiative in Kenya. To produce French beans, a water-intensive crop, a farmer must have access to water for irrigation [39]. There are only two types of irrigation infrastructure available in the community, which serve only 20% of the smallholder households, implying that only households within this group can exploit the opportunity provided by the initiative [39]. Membership to the two irrigation schemes comes at the smallholder’s own cost (Kshs. 66,000 and 120,000, equivalent to US $610 and $1110). These are funds that nearly all non-members (80%) insist they were unable to raise. In addition to water resource limitations, the non-participants have less than average farm plots sizes (1.65 acres) compared with participants (2.18 acres). An increase in farm plot size significantly increases chances of participating in the French bean business.

The trend of exclusion of the least well-off smallholders from the initiatives mentioned above seems to be replicated in the malt barley initiative in Ethiopia, the mango enterprise in Kenya, and the farm forestry business in South Asia. Worku [41] finds that smallholders engaging in malt barley production for Heineken in the Arsi Zone have significantly higher production resources (farm plot sizes of 4.9 acres for participants versus 3.7 acres for non-participants) and labor resources (family size of 6.8 for participants versus 6.1). Based on [42], participation in the malt barley initiative is dictated by farming resource endowment (farm plot size, size of livestock holding), information access, institutional linkage, gender, and location. In Makueni county in Kenya, the household’s farm plot size is a significant determinant for participation in the mango business; participants’ households’ average plot size is larger—12 acres—than non-participants—7 acres [40]. Participants also have a higher number of mango trees and larger household labor capacity (family sizes) than non-participants. The study from Tanzania [48] finds that participation in inclusive businesses is determined by assets (vehicle), access to information, age, livestock holding (for livestock business), level of non-farm income, distance to the urban area, and education level. In the farm forestry initiative in South Asia, ‘the farmers who participated in this business project were not the poorest, but rather middle-income farmers’ and consisted of those with bigger farm plot sizes [52].

The extensive review study on inclusive businesses across Asia and Africa by Kaminski et al. [49] corroborates the learnings from the cases presented above. As part of their study’s conclusion, they maintain that their finding ‘[…] suggests that most of these models require smallholders to have access to some degree of assets, such as land, finance and/or human and social capital’. German et al. [50] share similar findings. They note that ‘[…] factors specific to the crops and value chains reviewed also shape social inclusion’. They mention barriers to mechanization, heavy input costs, delayed returns, financial risks, and the need for water for irrigation. When the private sector is involved (and government support thus is scaled back), the support available to address these constraints is likely to be inadequate, given the competitiveness and efficiency imperative in the agri-food sector. A summary of these findings is presented in Table 2.

As a pro-poor strategy in smallholder communities, the inclusive business approach is designed to integrate poor and marginalized rural populations into local, regional, and global agri-food value chains as part of the Sustainable Development Goals’ call of ‘leaving no one behind’ [89]. While generally the benefits attributed to participation in IBMs seem to apply to some smallholders, it becomes a concern when those smallholders most in need are excluded from participation. This outcome defeats the logic behind inclusive business development—reaching those furthest behind first. Importantly, what does it mean for food and nutrition security in the respective communities? The next section attempts to answer this question.

### 3.3. Inclusive Businesses’ Contribution to Smallholder Household Food Security

The different initiatives analyzed provide a level of support needed to improve smallholder farming systems; hence, increased productivity and improved income can be expected among participant households. In Raya Azebo, Ethiopia, Gebru et al. [43] find that participation in the vegetable business significantly improves the income of the households involved. Accordingly, better income among participant households contributes to improved food security by enabling food access during lean periods, i.e., when they are short on food they have produced themselves. Further analysis, however, reveals that the increase in income does not have an effect on households’ food variety and dietary diversity, calorie intake, or child anthropometry. The authors attribute the outcome to the limited availability of diverse foods in local markets and possibly a lack of adequate knowledge of nutrition. Even more worrying, the findings indicated ‘a negative relationship between participation in the vegetable business and the scores for food variety and diet diversity’ [43].

The findings from the mango business in Makueni county in Kenya are similar to those from the Ethiopian vegetable business. The participants report a significant increase in their income, thanks to mango value addition and market opportunities not available to them previously [40]. The resultant enhanced income is associated with better general household food security. Nutrition-wise, however, no significant difference is observed between the business participant and non-participant households. The local diet, for both business participants and non-participants, greatly lacks in white roots and tubers, fruits, fish, eggs, and meat. The study indicates that rather than use the income from business to acquire better food, smallholders spend it on non-food items, such as education, healthcare, and agricultural inputs. In the event money is spent on food, it is to purchase staples and/or commercial ingredients such as oil, sugar, etc. Rarely, during special occasions such as holidays, do they spend income on ‘special food’ such as meat and eggs, absent in the local diet.

In the first malt barley initiative in Ethiopia, increased income is associated with better food access and diet diversity score, but does not improve ‘the actual calorie intakes, food variety score, child nutrition and food availability’ [42]. The findings from the second malt barley business in Ethiopia indicate a significantly better income for participant households [41]. The better income among participants is as a result of a premium price offered by Heineken and higher yields from improved seed varieties and other production support received from the business initiative. However, the increased income contributes neither to better household food security nor nutrition (dietary diversity). Much of the income from the malt barley business is spent on accumulating new assets instead of acquiring better and nutritious food. An additional explanation for this outcome is the company’s payment approach, where smallholders receive a lump sum instead of frequent periodic payments. Studies have shown that more frequent payments in contract farming arrangements favor household food security when compared to lump sum payments [90,91]. Should this be the case, the present outcome would barely change, given that, as reported by the author, there is a lack of certain food varieties, particularly fruits and vegetables, in the local market. In the Tanzania study, Herrmann et al. [48] find that despite significantly increasing business participant income compared to non-participants, no significant difference in household food security is observed. This study notes that other elements are critical to food security, including access to portable water, sanitation, healthcare, and education. In findings from the study on the six different IBM initiatives in Ethiopia, Tommasi [44] indicates that while there is limited contribution to food security, the dietary diversity among participant households does not improve.

In Tharaka Nithi county, Kenya, the French bean business is considered a failure, as it generally fails to have an impact on the participating households [39]. Owing to critical contextual factors, the crop performs extremely poorly. Despite availability of irrigation infrastructure among the business participants, access to water for crop production is both inconsistent and unreliable, because the community is in a semi-arid region where rain is erratic. As a result, crop failure is common. Out of all the households that engaged in the French bean production cycle during the time of the study, only 20% made some (limited) profit. A lack of improvement in participant household income implies that the business does not contribute to local food or nutrition security.

Regarding a direct contribution to household food availability from increased farm productivity, the outcome becomes a matter of whether the promoted crop is consumed locally. It is reported that the vegetable business in Ethiopia contributes to participants’ household food security via partial consumption of the produce [43]. The same applies to other initiatives that promote fruits and other food crops that form part of local diets. However, concerns arise when the crops are produced for the market and when income derived from the sale is not sufficient to procure food or the food needed is not available in local markets. Ntakyo and van den Berg [92] warn that ‘market-oriented crop production is not sufficient for reducing hunger and undernutrition of smallholder households, even if the marketable crop is a food crop that can also be consumed at home’. In the French bean business, the crop is produced primarily for export to European markets and is hardly consumed locally; thus it does not directly contribute to local food and nutrition security. Smallholders’ market orientation also implies land use change, risking a decline in local farms’ (food) crop diversity, common in smallholder households, which will in turn impact diverse food availability both in households and at local markets. In the case of Makueni county, the mango processing plant triggered smallholder interest in expanding the mango plantation in anticipation of increased returns from the crop [40]. To participate in the tabasco chili initiative in Zimbabwe, farmers use up to 0.25 ha (0.6 acres), in a community where the average farm is between 0.5 ha (1.2 acres) and 1. ha (2.5 acres). This means that, for some farmers, at least half of the farm is allocated to the cash crop [47]. The average income in a year from the business is US $260 ($0.7 per day), which does not help participants beat the poverty line threshold of US $1.90 per day. Such an amount does not contribute to income enough to ensure smallholder food security, especially given the other competing non-food needs. Considering how small the available land is, tabasco chili production is also likely to reduce the diversity of food crops. Studies have shown that such reduced crop diversity in smallholder farms is detrimental to household dietary diversity [45,93,94].

Having been excluded, non-participant households do not benefit from the businesses’ pathways to improved food security—increased farm productivity and income. Hence, the potential food security benefits cannot be expected for this group of smallholders. In very few cases can benefits to non-participants be seen; these include indirect benefits through employment and trade in the produce, as reported in various cases [40,43,44,46,47]. Based on the six IBM initiatives in Ethiopia, it is reported that the income non-participants derive from the employment opportunities created is insufficient to contribute to better local food security. The income, and by extension, food security benefits, are marginal, and hence have not been reviewed. What is clear is that in all the businesses, the excluded smallholders fair worse in their household food security than those included. With regard to nutrition security, however, neither participants nor non-participants saw improvements. This implies a significant nutrition gap that goes beyond the farmers’ production and marketing capacities. The unavailability of diverse food items in the local markets, the competition of food needs with non-food needs for the available income, and limited nutrition knowledge, among other reasons, need to be acknowledged. Table 3 presents an overview of the food security contribution of different inclusive business initiatives.

## 4. Discussion and Conclusions

The present paper, adopting a food systems approach, set out to broaden the knowledge of and insight into the contribution of inclusive business to food and nutrition security in smallholder communities. This paper serves to enrich the existing knowledge and debate the role of the private sector in delivering societal development goals, specifically food security, for marginalized populations [2,5,89,95,96,97,98]. It does so by expanding the discussion on the possibilities and improbability of inclusive businesses in solving smallholder-farming-related livelihood constraints, especially those most marginalized socio-economically and therefore most in need [8,99]. Concomitantly, and most importantly, it illuminates the importance of a food systems’ approach, which puts food at the center of policies and development interventions that are designed and implemented with a view to addressing food and nutrition insecurity in smallholder communities.

Today’s food security policies and interventions in developing countries favor investing in smallholder commercialization through an inclusive business approach [5,88,100,101,102,103]. Improving smallholder farms’ productivity (food availability) and income (food accessibility) are among the key elements of an effective smallholder food system [37]. In the context of all the studies presented, these outcomes seem to only apply to a proportion of smallholders, the relatively well-off, and apply only to a certain extent, depending on the initiative, owing to diversity of the socio-economic characteristics in the respective communities. The individual smallholders’ resources, such as farm plot size, labor, and access to water for irrigation, determine participation in the inclusive business initiatives. Land and water resources are particularly critical. In smallholder communities, they constitute the basis through which livelihoods are earned. Therefore, they are key ingredients to secure households’ food needs. As such, the individual smallholders’ quality of land and water resources are central to their food system.

The exclusion of smallholders that are most in need, irrespective of the support provided through inclusive businesses or instances where an initiative fails to meet its basic goals as in the case of the French bean project in Kenya, implies the approach may not be realistic in certain smallholder settings. While the limitations of inclusive business models both in achieving the desired outcome and reaching the marginalized may be a result of poor design and implementation of an initiative, the capacity issue of the business models should also be acknowledged. Considering that inclusive businesses are profit oriented, it is understandable that not every smallholder can be reached, particularly those with very low production capacity, as doing so would raise the transaction costs, rendering the businesses unprofitable and unsustainable. As pointed out by van Westen et al. [8], like any other business, inclusive business models ‘work within bounded rationalities and market pressures’. Against this background, inclusive agribusinesses that ‘integrate farmers into capital intensive markets’ may not be the ‘best solution’ to addressing smallholders’ livelihood issues [49]. The initiatives are promoted with the view of enhancing local food security; the exclusion of most food insecure households without an alternative strategy implies a major shortcoming in policies supporting such initiatives and can perpetuate and even exacerbate existing inequalities.

Based on the present review, the policies and interventions that promote inclusive business models seem not to consider the significant variation in the local structural factors—land, water, and other constraints—that determine smallholder livelihoods and food security. This calls for rethinking the application of the inclusive business model in smallholder communities, particularly with regard to a clear indication of to whom it applies. Perhaps some of the constraints that prevent smallholders’ participation in the present inclusive business models, for instance, access to water for irrigation and cooperative membership, could be addressed by an increase in the investments from donors and the private sector. For those most constrained in production resources, nearly landless smallholders for instance, alternative interventions should be considered. For example, provision of near free inputs and technical knowledge for one’s own food production, as a social safety net for the poorest and most marginalized with some access to land, may be a useful way to support this category of smallholders to enhance their food security. After all, most of the food consumed in the smallholder households comes from their own farms. Interventions that support non-farm income also present an important avenue for improving livelihoods and food security in smallholder communities. The Raya Azebo case, where non-participants benefit from the business initiative through vegetable trade and youth employment in Makueni businesses, provides vital evidence for the possibility of non-farm contribution to local livelihoods. Studies show that nonfarm income, such as wages from labor in other farms or other sources, is a significant part of rural households’ income, an its relevance increases with the growing population and dwindling per capita agricultural resources [104,105,106,107,108]. The non-farm income must be adequate (living wage) to meet people’s needs. For the most vulnerable, who cannot adequately benefit from farming and/or non-farming interventions, social protection programs and safety net plans [109,110] are necessary in the fight against hunger and malnutrition.

The failure of inclusive businesses to contribute to household dietary diversity is a major concern, particularly when an increase in the farm income does not necessarily improve nutritious food intake. This finding highlights a critical contradiction to the expected outcome, which is the basis for agribusiness promotion as a solution to smallholder food and nutrition security. The studies demonstrate that smallholders have needs beyond food that must be met from the income raised. These socio-economic aspects become important elements for a smallholder food system, in that they influence whether additional income can be spent on the household’s food needs. The diversion of the business participants’ income to non-food expenditures, as seen in the mango (Kenya) and malt barley (Ethiopia) initiatives, shows that even with extra support, the income derived from the business is not sufficient to meet the dietary needs for the participant households. To understand these local dynamics and how they influence smallholders’ household food and nutrition security, one must look outside the value chain effects on farm productivity and income. Specifically, the level of improvement in household income, and how individual households prioritize different households’ needs against the available income/resources, need to be investigated. This points to a mismatch between the logic and assumption of policymakers and practitioners, the quality of the interventions’ impact, and the decisions of the targeted smallholders.

In summary, there is a need for policymakers, donors, development agencies, and businesses pursuing inclusive business approaches to broaden the scope by promoting complementary interventions in the context of a food systems perspective. While there are limitations on how to conceptualize and apply the concept of food systems, it certainly adds significant depth to the understanding and delivery of interventions meant to improve smallholder food security. Inclusive business approaches in smallholder agriculture focus on changing household farm productivity and income prospects. Although this is an important step towards creating opportunities for improved household food and nutrition security, it is evident that many aspects, including those that relate to diversity in production resource capacity, are not acknowledged. Importantly, the socio-economics factors that surround food systems, particularly outside the agribusiness value chains, seem to be overlooked. Depending on the local context for smallholders, engaging in commercial farming—which is often resource and cost-intensive—can result in the use of household savings and finances reserved for acquiring better food or the fact that the income obtained is met with many different priorities. Also, there is the issue of time lost in time-intensive cash cropping at the expense of searching for and preparing food. Understanding these socio-economic dynamics of smallholder food systems and responding to them appropriately requires a broader approach. This paper demonstrates that a food systems approach, which demands a multidisciplinary approach by ensuring increased integration of actors from different disciplines—agri-ecology, nutrition, economics and sociology—offers such a platform [68,111]. Local food systems and their challenges vary in different communities and require context-specific approaches. A food systems approach provides for a more nuanced understanding of the critical issues to improve specific food systems’ elements in delivering food and nutrition to any individual in any community.

## Figures and Tables

**Figure 1 foods-10-01785-f001:**
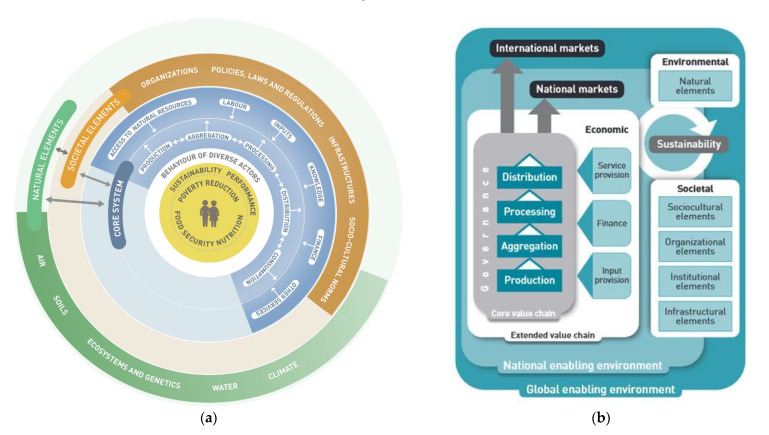
(**a**) Sustainable food systems framework [66]. (**b**) Sustainable value chain framework [67].

**Table 1 foods-10-01785-t001:** Material included in the analysis.

Country	Author(s) and Year	Primary Inclusive Features	Produce
Ethiopia	Gebru et al., 2019 [42]	Production/marketing support	Malt barley
Ethiopia	Gebru et al., 2019 [43]	Production/marketing support	Vegetables
Ethiopia	Worku 2019 [41]	Production/marketing support	Malt barley
Ethiopia	Tommasi 2018 [44]	Production/marketing support	Various: Livestock/dairy; Food/cash crops
Ghana	Mangnus and van Westen 2018 [45]	Production/marketing support	Maize
Ghana	de Veries 2017 [46]	Production/marketing support	Various: Fruit crops
Kenya	Wangu et al., 2021 [39]	Production/marketing support	French bean
Kenya	Wangu et al., 2020 [40]	Production/marketing support	Mango
Zimbabwe	Mutema and Chiromo 2014 [47]	Production/marketing support	Tabasco chili
Tanzania	Herrmann et al., 2018 [48]	Production/marketing support	Various: Food and cash crops
Various	Kaminski et al., 2020 [49] (Review article)	Production/marketing support	Aquaculture
Various	German et al., 2020 [50] (Review article)	Production/marketing support	Various: Food and cash crops
Various	World Bank 2018; 2012 [51,52] (Report)	Production/marketing support	Various

**Table 2 foods-10-01785-t002:** Participation in inclusive business initiatives.

Initiative	Factors Determining Participation in IBMs
Vegetable (Ethiopia)	Productive resources (farm plot size, irrigation infrastructure), cooperative membership, extension services, age of household head, market distance, risk perception
Malt barley I (Ethiopia)	Production resources (farm plot size, size of livestock holding, labor), information access (radio), age, institutional support, distance to market, risk perception
Malt barley II (Ethiopia)	Production resources (land plot size, labor)
French bean (Kenya)	Productive resources (farm plot size, irrigation infrastructure)
Mango (Kenya)	Productive resources (farm plot size, number of mango trees, labor), access to loans, age, education of household head
Food and cash crops (Tanzania)	Production resource/asset (size of livestock holding, vehicle), distance to market, non-farm income, information access
Farm forestry (South Asia)	Production resources (farm plot size)
Aquaculture (developing countries)	Production resources (farm plot size), finance, human and social capital
Food and cash crops (developing countries)	Production resources (irrigation, machinery) risk perception

**Table 3 foods-10-01785-t003:** Inclusive businesses’ contribution to participants’ household food security.

Initiative	Food Security Contribution (and Reasons/Factors Where Applicable)
Vegetable (Ethiopia)	- Improved income leads to better food security - No dietary change (participants/non-participants) or caloric intake or child anthropometry improvement - Limited availability of food in local markets and nutrition knowledge among contributing factors
Malt barley I (Ethiopia)	- Improved income leads to better food security - No improvement in calorie intake, food variety, child nutrition, or food availability
Malt barley II (Ethiopia)	- Improved income does not lead to better food or nutrition security - Income spent on acquiring new assets
Various (Ethiopia)	- Improved income contributes to limited food security - No contribution to dietary diversity
French bean (Kenya)	- No income improvement; does not affect food or nutrition security - Mismatch between intervention and local context - Crop produced not part of local diet
Mango (Kenya)	- Income improved, associated with better food security - No dietary diversity improvement - Income spent on non-food needs - Potential reduction in crop diversity
Maize (Ghana)	- Increased income but limited long-term contribution to food security - Reduce food crop diversity
Fruits (Ghana)	- Improved income leads to limited contribution to food security - No dietary diversity contribution
Tabasco chili (Zimbabwe)	- Limited income improvement; little contribution to food security - Likely to reduce food crop diversity
Food and cash crops (Tanzania)	- Improved income does not lead to better food or nutrition security - Access to other elements, such as portable water, sanitation, healthcare, and education, also influences food security

## Data Availability

All the materials reviewed in this paper are available upon request from the author.

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
