# Peer review of "The Need for a Food Systems Approach in Smallholder Food and Nutrition Security Initiatives: Lessons from Inclusive Agribusiness in Smallholder Communities"

_foods, 2021, doi:10.3390/foods10081785_

Round 1
Reviewer 1 Report
This paper focuses on the highly relevant topic of how to improve livelihoods of small scale farmers and whether inclusive business models may serve this pressing task. The argumentation of the paper is convincing on a general level and the analysis is plausible also on a more detailed level. The paper would clearly benefit from graphical representations of key concepts such as food systems, and of crucial contrasting concepts such as food value chains. Moreover, it should summarize key findings in the form of tables. The chapter structure and chapter titles could be improved, e.g., the section on conclusions is not distinguished. The chapter “The scope of inclusive business models”, to take another example, should be introduced beforehand (maybe at the end of the previous chapter) by some information how the author is going to proceed and what the reader can expect from the chapter.
My most serious concern relates to methodology. As I understand it, the author does not clarify how he selected the case studies and reviews that are the empirical basis for the evaluation. Moreover, it remains unclear to me why the paper only includes four case studies and how these relate to the reviews that are also included. This raises even more questions, in my point of view, as the author – if I understand correctly – is in fact also the author of two of the case studies that are part of the assessment. If the author of the paper has in fact published two of the case studies that he is referring to, I am wondering why he is not exploiting these case studies in greater depth, since he has then the opportunity to examine them in the food systems perspective that he advises. Also he should indicate his participation in the said case studies. One final point in this matter: the empirical basis of the included reviews remains unclear in the paper. A table etc. shoud indicate how cases were selected for the cited reviews, how many cases were analyzed etc.
Resuming the critical issues in the current version of the paper, I suggest a major revision. It should clarify the issues raised by adding more information, and I strongly recommend that a more transparent and general overview of the available studies on inclusive business models is given, i.e., by summarizing main findings etc., and that the selection of case studies that the paper is analyzing or re-analyzing is justified in a transparent way. Actually I recommend to include more case studies in the analysis. Of course, their selection should be justified by transparent criteria (testing a specific hypothesis, including the broadest range of variation in variable X, investigating the possible influence of condition Y on outcome Z etc.), and the procedure of selection (how the corpus is formed, with which keywords etc.) must be clarified.
In turn, the introduction and conceptual background can be shortened, in my point of view. The paper should more directly proceed to the core questions and concepts that it addresses and applies, possibly enhanced by tables and diagrams. The introduction should already present the methodology of the paper, the key research question and the key findings in a nutshell.
The language must be improved, for there are not only grammatical errors and stylistic flaws, but also some expressions that do not make sense and could be in part the result of insufficient attention. For instance, in line 53, p 2, the verb is missing: “Participation in the businesses market-oriented production instead of for home consumption.” Another example is line 85, p. 2, where it is unclear to me to what “system” refers in the citation: “ ("System", 2020; Halberg and Westhoek, 2019; Grant, 2015) 85 (Halberg and Westhoek, 2019; Grant, 2015)”. In line 93, p. 2, a section seems to be quoted, but it is unclear where the quotation ends, see “...as van Berkum et al. (2018: 2) indicate, ‘a food systems approach is a useful interdisciplinary conceptual framework for research and policy aimed at sustainable solutions for the sufficient supply of healthy food. Through the food systems’ perspective, the delivery of food is viewed as a complex interaction of sub-systems with feedback loop than a simple linear process (IPES-Food, 2019; van Berkum et 97 al., 2018; UNEP, 2016).” The acronym FNS must be explained. Sometimes expressions lack in clarity, see e.g., line 264, p. 6: “Conversely, the present paper takes a food systems approach in understanding the value chains – inclusive business models – contribution to smallholders’ household food security.” This sentence raises the question whether a value chain is the same as an “inclusive business model”? To take another example, line 288, p. 6 is unclear to me: “The former research includes an initiative that was carried out by the International Financial cooperation to improve smallholder livelihoods in Asia, and two extensive reviews on inclusive business initiatives in various developing countries - (Kaminski et al., 2020; German et al., 2020; WB, 2018; WB, 2012).” First, four publications are cited, not two, secondly, the whole paragraph is unclear about what the selection criteria of the studies have been: initiatives for inclusive business models or also initiatives that only focused on local food security, in which case the sampling of studies seems at odds with the aim of the evaluation.
This paper could be a significant contribution to a very topical discussion and in view of improving smallholders’ livelihoods. In this spirit I am hoping for a corresponding improvement of this promising paper!
Author Response
Reviewer 1 comments and response (in bold)
Comments and Suggestions for Authors
This paper focuses on the highly relevant topic of how to improve livelihoods of small scale farmers and whether inclusive business models may serve this pressing task. The argumentation of the paper is convincing on a general level and the analysis is plausible also on a more detailed level. The paper would clearly benefit from graphical representations of key concepts such as food systems, and of crucial contrasting concepts such as food value chains. Moreover, it should summarize key findings in the form of tables. The chapter structure and chapter titles could be improved, e.g., the section on conclusions is not distinguished. The chapter “The scope of inclusive business models”, to take another example, should be introduced beforehand (maybe at the end of the previous chapter) by some information how the author is going to proceed and what the reader can expect from the chapter.
I have done a major re-write on some section as the track changes shows for better presentation of the paper:
- Included contrasting information on main concept as well as what they have in common, including graphical representations as suggested.
- Summary of key findings in form of tables
- More sub-heading clarity, including distinguishing discussion and conclusion from findings section
- Clear introduction of sub-sections
My most serious concern relates to methodology. As I understand it, the author does not clarify how he selected the case studies and reviews that are the empirical basis for the evaluation. Moreover, it remains unclear to me why the paper only includes four case studies and how these relate to the reviews that are also included. This raises even more questions, in my point of view, as the author – if I understand correctly – is in fact also the author of two of the case studies that are part of the assessment. If the author of the paper has in fact published two of the case studies that he is referring to, I am wondering why he is not exploiting these case studies in greater depth, since he has then the opportunity to examine them in the food systems perspective that he advises. Also he should indicate his participation in the said case studies. One final point in this matter: the empirical basis of the included reviews remains unclear in the paper. A table etc. shoud indicate how cases were selected for the cited reviews, how many cases were analyzed etc.
As suggested, I have re-visiting the methodology and research design; which I made part of the introduction (included clear questions and motivation and approach I used for the selection of the cases). Cases included are now also presented in form of a table.
Resuming the critical issues in the current version of the paper, I suggest a major revision. It should clarify the issues raised by adding more information, and I strongly recommend that a more transparent and general overview of the available studies on inclusive business models is given, i.e., by summarizing main findings etc., and that the selection of case studies that the paper is analyzing or re-analyzing is justified in a transparent way. Actually I recommend to include more case studies in the analysis. Of course, their selection should be justified by transparent criteria (testing a specific hypothesis, including the broadest range of variation in variable X, investigating the possible influence of condition Y on outcome Z etc.), and the procedure of selection (how the corpus is formed, with which keywords etc.) must be clarified.
In turn, the introduction and conceptual background can be shortened, in my point of view. The paper should more directly proceed to the core questions and concepts that it addresses and applies, possibly enhanced by tables and diagrams. The introduction should already present the methodology of the paper, the key research question and the key findings in a nutshell.
This has been addressed by including two questions that the study seeks to answer, as well as clear reasons why the study focuses on the selected studies.
While I do not include key findings in the introduction, I elaborate on the reason why the study focuses on the said questions and how these questions fit within the overall goal of the study.
The language must be improved, for there are not only grammatical errors and stylistic flaws, but also some expressions that do not make sense and could be in part the result of insufficient attention. For instance, in line 53, p 2, the verb is missing: “Participation in the businesses market-oriented production instead of for home consumption.” Another example is line 85, p. 2, where it is unclear to me to what “system” refers in the citation: “ ("System", 2020; Halberg and Westhoek, 2019; Grant, 2015) 85 (Halberg and Westhoek, 2019; Grant, 2015)”. In line 93, p. 2, a section seems to be quoted, but it is unclear where the quotation ends, see “...as van Berkum et al. (2018: 2) indicate, ‘a food systems approach is a useful interdisciplinary conceptual framework for research and policy aimed at sustainable solutions for the sufficient supply of healthy food. Through the food systems’ perspective, the delivery of food is viewed as a complex interaction of sub-systems with feedback loop than a simple linear process (IPES-Food, 2019; van Berkum et 97 al., 2018; UNEP, 2016).” The acronym FNS must be explained. Sometimes expressions lack in clarity, see e.g., line 264, p. 6: “Conversely, the present paper takes a food systems approach in understanding the value chains – inclusive business models – contribution to smallholders’ household food security.” This sentence raises the question whether a value chain is the same as an “inclusive business model”? To take another example, line 288, p. 6 is unclear to me: “The former research includes an initiative that was carried out by the International Financial cooperation to improve smallholder livelihoods in Asia, and two extensive reviews on inclusive business initiatives in various developing countries - (Kaminski et al., 2020; German et al., 2020; WB, 2018; WB, 2012).” First, four publications are cited, not two, secondly, the whole paragraph is unclear about what the selection criteria of the studies have been: initiatives for inclusive business models or also initiatives that only focused on local food security, in which case the sampling of studies seems at odds with the aim of the evaluation.
The paper has undergone extensive English proof to ensure the grammatical errors, stylistic flaws and awkward phrasing have been addressed. The issues regarding the cases included and criteria have also been effectively addressed.
This paper could be a significant contribution to a very topical discussion and in view of improving smallholders’ livelihoods. In this spirit I am hoping for a corresponding improvement of this promising paper!
Thank you for reviewing the paper; the comments have been very useful for improving on the previous draft.

Reviewer 2 Report
This article contains excessive repetition of the conceptual limitations of the "inclusive business" approach to providing nutritional food security and the potential advantages of a "food systems" approach to ensuring nutritional food security. While the review of existing literature clearly documents the limitations of the inclusive business approach, no comparable review of literature is provided that supports the superiority of the food systems approach. While there is some explanation of why food systems approach is needed and what factors and measures should be included, there is little evidence that a food systems approach has been documented to be superior to the inclusive business approach. It seems clear from the previous research reviewed that the inclusive business approach is inadequate, perhaps even inappropriate, in assessing changes in food security. This is a very important conclusion and potentially valuable information for any institution or organization that is engaged is activities related to food security, community development, or agri-food sustainability. Perhaps a better title for an edited version of this article would be something like "The Need for a Food Systems Approach to Nutritional Food Security." Such a title would not lead the reader to expect a review of studies that document the superiority of the food systems approach monitoring food security.
Author Response
Reviewer 2 comments and response (in bold)
Comments and Suggestions for Authors
This article contains excessive repetition of the conceptual limitations of the "inclusive business" approach to providing nutritional food security and the potential advantages of a "food systems" approach to ensuring nutritional food security. While the review of existing literature clearly documents the limitations of the inclusive business approach, no comparable review of literature is provided that supports the superiority of the food systems approach. While there is some explanation of why food systems approach is needed and what factors and measures should be included, there is little evidence that a food systems approach has been documented to be superior to the inclusive business approach. It seems clear from the previous research reviewed that the inclusive business approach is inadequate, perhaps even inappropriate, in assessing changes in food security. This is a very important conclusion and potentially valuable information for any institution or organization that is engaged is activities related to food security, community development, or agri-food sustainability. Perhaps a better title for an edited version of this article would be something like "The Need for a Food Systems Approach to Nutritional Food Security." Such a title would not lead the reader to expect a review of studies that document the superiority of the food systems approach monitoring food security.
The manuscript has undergone a major re-write to address your comments/suggestions.
- First, an extensive English proof has been conducted to ensure the grammatical errors, stylistic flaws and awkward phrasing have been addressed
- The overlapping content has been addressed
- I have included a comparative text on the food systems approach and value chains approach (commonly used in inclusive business) (in the first paragraph of the conceptual background, it is clear why food systems approach is better for analyzing food insecurity– as reviewer 1 suggested, I have even included a graphical representation of the two concepts/approaches
- Based on your feedback, I have improved the title of the paper
Thank you for reviewing the paper; the comments have been very useful for improving on the previous draft

Round 2
Reviewer 2 Report
Major revisions in presentation of this manuscript have resulted in a major improvement in its overall quality and merit. The relevance and importance of the case studies in reaffirming conclusions from the review of research are much clearer than in the original manuscript. The basic conclusions of this study provide guidance for future research and are of potential importance in guiding public policies to address food insecurity.